# Predicting the Beneficial Effects of Cognitive Stimulation and Transcranial Direct Current Stimulation in Amnestic Mild Cognitive Impairment with Clinical, Inflammation, and Human Microglia Exposed to Serum as Potential Markers: A Double-Blind Placebo-Controlled Randomized Clinical Trial

**DOI:** 10.3390/ijms26041754

**Published:** 2025-02-19

**Authors:** Ruth Alcalá-Lozano, Rocio Carmona-Hernández, Ana Gabriela Ocampo-Romero, Adriana Leticia Sosa-Millán, Erik Daniel Morelos-Santana, Diana Zapata Abarca, Dana Vianey Castro-de-Aquino, Edith Araceli Cabrera-Muñoz, Gerardo Bernabé Ramírez-Rodríguez, Ana Luisa Sosa Ortiz, Eduardo A. Garza-Villarreal, Ricardo Saracco-Alvarez, Jorge Julio González Olvera

**Affiliations:** 1Laboratorio de Neuromodulación, Subdirección de Investigaciones Clínicas, Instituto Nacional de Psiquiatría “Ramón de la Fuente Muñiz” (INPRFM), Mexico City 14370, Mexico; 2División de Estudios de Posgrado, Facultad de Medicina, Programa de Ciencias Médicas, Odontológicas y de la Salud, Universidad Nacional Autónoma de México (UNAM), Mexico City 04510, Mexico; 3Dirección de Servicios Clínicos, Instituto Nacional de Psiquiatría “Ramón de la Fuente Muñiz” (INPRFM), Mexico City 14370, Mexico; 4Laboratorio de Neurogénesis, Subdirección de Investigaciones Clínicas, Instituto Nacional de Psiquiatría “Ramón de la Fuente Muñiz” (INPRFM), Mexico City 14370, Mexicoedarcamu@yahoo.com.mx (E.A.C.-M.); 5Laboratorio de Demencias, Instituto Nacional de Neurología y Neurocirugía Manuel Velasco (INNN), Mexico City 14269, Mexico; 6Instituto de Neurobiología, Universidad Nacional Autónoma de México Campus Juriquilla, Querétaro 76230, Mexico; 7Dirección de Investigaciones en Neurociencias, Instituto Nacional de Psiquiatría “Ramón de la Fuente Muñiz” (INPRFM), Mexico City 14370, Mexico; 8Comisión Nacional Contra las Adicciones, Mexico City 10200, Mexico

**Keywords:** amnestic mild cognitive impairment, mild cognitive impairment, transcranial direct current stimulation, non-invasive brain stimulation, cognitive stimulation, cognition, microglia, interleukin-6, BDNF, CX3CL1

## Abstract

In amnestic mild cognitive impairment (aMCI), neuroinflammation evolves during disease progression, affecting microglial function and potentially accelerating the pathological process. Currently, no effective treatment exists, leading to explorations of various symptomatic approaches, though few target the underlying physiological mechanisms. Modulating inflammatory processes may be critical in slowing disease progression. Cognitive stimulation (CS) and transcranial direct current stimulation (tDCS) applied to the left dorsolateral prefrontal cortex (l-DLPFC) show promise, but the results are heterogeneous. Thus, a randomized, double-blind, placebo-controlled clinical trial is currently underway. The first-stage results were examined after three weeks of intervention in two groups: active tDCS combined with CS and sham tDCS combined with CS. Twenty-two participants underwent two assessments: T0 (baseline) and T1 (after 15 sessions of tDCS, active or sham, and 9 sessions of CS). The results demonstrated that CS improved cognition, increased brain-derived neurotrophic factor (BDNF) levels, and reduced peripheral proinflammatory cytokine levels (interleukin IL-6 and chemokine CX3CL1) in serum. This decrease in IL-6 may promote microglial proliferation and survival as a modulatory effect response, while the increase in BDNF might suggest a regulatory mechanism in microglia–neuron interaction responses. However, tDCS did not enhance the cognitive or modulatory effects of CS, suggesting that longer interventions might be required to achieve substantial benefits.

## 1. Introduction

Mild cognitive impairment (MCI) represents a critical stage in the progression toward Alzheimer’s disease (AD) [1]. This condition is characterized by an abnormal decline in cognitive function compared to normal aging [2], with memory deficits being the most common form. Amnestic MCI (aMCI) is particularly concerning, as it carries a higher risk of progressing into AD, with a 15% probability of conversion within two years [3]. Non-amnestic MCI affects other cognitive domains and is linked to a higher risk of non-AD dementia [4,5,6], while multi-domain MCI impacts both memory and other cognitive functions [2,3]. Despite advances in the development of treatments for MCI, strategies are urgently needed to address the underlying pathophysiological processes and the symptomatic aspects of the condition, ideally supported by biomarkers that can confirm disease-modifying effects [7,8,9].

While age is a significant risk factor for the development of MCI, several modifiable factors offer opportunities to prevent or delay the progression to dementia [10,11,12]. In Mexico, studies suggest that up to 56% of the population may benefit from interventions targeting these modifiable risk factors [10,13,14]. Despite the potential for intervention, there are no treatments approved for MCI by international health regulatory agencies due to the little positive impact demonstrated in MCI trials [1,7,15,16]. Thus, non-pharmacological interventions are gaining attention, particularly those that target the ability of the brain to adapt and reorganize [17].

During MCI, the prefrontal cortex (PFC) activates compensatory mechanisms driven by neuroplasticity, which can be enhanced to improve the affected neural networks [18]. This phenomenon, linked to cognitive reserve, underpins therapeutic approaches aimed at optimizing the affected brain circuits [19] and slowing down the progression toward dementia [10,14,17]. In this regard, cognitive stimulation (CS) is an intervention involving active engagement of cognitive skills and strategies to obtain new information, often in a group setting [20]. Meanwhile, transcranial direct current stimulation (tDCS), a non-invasive brain stimulation (NIBS) technique, modulates brain network activity and holds promise as a therapeutic approach to improve cognitive processes [21,22]. In this regard, anodal stimulation enhances cognitive function and modulates cortical excitability, promoting neuroplasticity [23]. Although it does not directly counteract the neuropathological changes associated with MCI, it may facilitate Aβ clearance through a potential vascular mechanism [24,25]. Furthermore, regular and relatively prolonged tDCS use significantly increases regional cerebral metabolism in MCI patients [26]. Regarding the duration of tDCS effects, these may also persist beyond the intervention period, strengthening connectivity between remote but functionally interconnected brain areas, particularly when combined with cognitive tasks [27]. However, the current evidence on the impact of tDCS on the progression rate from MCI to dementia remains limited and inconclusive. Regarding CS, 90 min group sessions held three times a week over 12 weeks showed significant improvements in pre- and post-intervention test scores [28]. The benefits of the social environment for MCI were highlighted compared to home-based cognitive interventions [20]. Notably, combining CS with tDCS may enhance cognitive performance in individuals with the possibility of modifying the progression of MCI [26,29,30]. Furthermore, “online” computerized CS and anodic tDCS applied to the left dorsolateral prefrontal cortex (l-DLPFC) are complementary, potentially leading to improved cognitive outcomes [20,31,32].

However, preliminary studies combining low-recurring interventions or a few tDCS and CS sessions have yielded mixed results [33,34]. While some studies report positive effects on cognitive tasks [31,32], others show no significant improvements [35,36,37,38]. This variability highlights the need for more research to determine the conditions under which these interventions are applied. Given that neuroplastic changes induced by tDCS are dose-dependent [39], it is crucial to fine-tune session parameters, including repetition, frequency, number of sessions, and stimulation modalities, to maximize the observed benefits [33,34]. This includes even less studied modalities, such as offline tDCS with group CS, and approaches that explore the modulatory effects following interventions [40,41].

In addition to cognitive interventions, addressing inflammation is increasingly recognized as a key strategy for managing MCI and AD. Proinflammatory markers such as interleukin-6 (IL-6) and the CX3CL1, also known as fractalkine (FKN), and brain derived neurotrophic factor (BDNF) have been linked to the pathophysiology of both MCI and AD [42,43]. These biomarkers, particularly IL-6 and CX3CL1, may provide insights into the neurodegenerative process, as their elevated levels indicate neuroinflammation, which is associated with cognitive decline [43,44]. Moreover, reduced levels of BDNF, a key factor for memory and cognitive function, have been observed in individuals with MCI [45,46]. Understanding the interaction between these inflammatory markers and BDNF, alongside microglial activity -which has a dual role in neurodegeneration, either protecting or exacerbating neuronal damage depending on the inflammatory environment- could provide valuable insights [47]. This understanding may help clarify how interventions like CS and tDCS influence both the molecular factors that modulate the effects and the cognitive aspects of MCI.

Thus, in the first stage of our study, we aimed to address this gap by asking whether the combined effects of CS and tDCS can improve cognitive performance and modulate biological parameters related to inflammation and microglia exposed to serum in individuals with aMCI, as the primary objective. Specifically, we assessed the effects on IL-6, CX3CL1, and BDNF levels, as well as on microglial viability and proliferation, to explore the underlying cellular and molecular mechanisms induced by these interventions. As a secondary objective, we determined whether a relationship exists between the distribution patterns of clinical, cognitive, and biological parameters within the intervention groups and identified which variables contributed most to explaining the variance.

## 2. Results

### 2.1. Demographic and Clinical Characteristics

From June 2024 to September 2024, 157 patients were recruited to consider their probable entry into the study. Of the total number of patients evaluated, 139 patients were excluded according to the selection criteria mentioned in the materials and methods section. The reasons for exclusion are presented in Appendix A. The 22 patients included in the study were randomized to receive one of the two intervention modalities: active tDCS plus CS or sham tDCS plus CS. Due to the balancing of the randomization, there are currently 14 sham and 8 active participants. Matching selection was carried out between active and sham participants based on similar characteristics. Four sham patients were discarded, resulting in a final selection of ten sham and eight active participants, being eighteen patients in total. The intervention was generally well tolerated, with only transient adverse effects such as uncomfortable scalp sensations or nausea following tDCS application. Meanwhile, the CS program was offered to the entire sample without any problems to report. Despite the expected difficulties at the beginning of each group, they were able to integrate, and each participant reported having achieved individual benefits from the interaction. The groups were similar in demographic data and baseline results. In both the active treatment and sham treatment groups, the sample was predominantly female (active = 75%; sham = 80%), with an average age range of 65 to 73 years, and 7 to 18 years of schooling. Regarding the onset and progression of the disease, it was found that the onset was insidious, and the progression was gradual in both groups. The data in Table 1 indicate that, based on the *p*-values, there are no significant differences between the groups in terms of demographic and clinical characteristics. Additionally, Appendix A present descriptive statistics of risk factors that may complicate the progression of MCI.

In cognitive parameters, significant main effects of time were observed in both groups after the intervention, with a large effect size in total Montreal Cognitive Assessment (total MoCA) (F = 4.8, *p* = 0.04, η^2^ = 0.23) and Cognitive Global Impression Scale–Severity (CGI-S) (F = 14.2, *p* = 0.001, η^2^ = 0.004). In the delayed verbal learning domain of the Screen for Cognitive Impairment in Psychiatry, a significant main effect of the group was found (F = 3.7, *p* = 0,07, η^2^ = 1.9). Additionally, a trend was suggested for the main effect of time in MoCA Memory Domain Score (MoCA Mem) (F = 3.4, *p* = 0.08, η^2^ = 0.17), Memory Failures of Everyday Questionnaire (MFEQ) (F = 3.6, *p* = 0.07, η^2^ = 0.18) and for the main effect of time in verbal fluency and processing speed domains of the SCIP-S (F = 3.8, *p* = 0.06, η^2^ = 0.19) (Table 2 and Figure 1). Additionally, Appendix A provide means and standard deviations for the outcome measures across groups and time.

### 2.2. Cognitive Stimulation Influences the Concentrations of IL-6 and CX3CL1

Biological factors related to inflammation are increased in the aMCI [48]. The concentrations of IL-6 in the serum of aMCI participants were quantified before and after the interventions (Table 3 and Figure 2A). For IL-6, a significant difference was found in the simple effect of time (F = 29.20, *p* = 0.00, η^2^ = 0.64), with significant decreases in both groups. Similarly, CX3CL1 also showed a significant reduction over time (F = 7.06, *p* = 0.01, η^2^ = 0.3). In both cases, no interaction was observed between the factors.

### 2.3. aMCI Patients Show an Increase in BDNF Levels Following the Intervention with CS

Several reports have indicated decreased BDNF levels in MCI patients [45,49,50,51]. Again, for BDNF (Table 3 and Figure 2A), a significant difference was found in the simple effect of time (F = 96.48, *p* = 0.00, η^2^ = 0.85), showing a significant increase in both groups after the intervention.

### 2.4. Serum of aMCI Patients Affects the Viability and Proliferation of Human Microglial Cells

Previous studies posited that peripheral proinflammatory cytokines may affect the response of microglial cells, which alters the activity of neurons [52]. Thus, we assessed the effect of soluble factors found in the serum of aMCI patients on human microglia (HMC3) cultured in different oxygen tension conditions (Figure 2B). For microglial cells incubated with 3% oxygen, we found a significant interaction of group and time (*p* = 0.013, F = 7.63, η^2^ = 0.32) with increased proliferation. However, in microglial cells cultured with 20% oxygen, a significant difference was found in the simple effect of time (*p* = 0.00, F = 115, η^2^ = 0.87). For cell viability on microglial cells cultured in 3 or 20% oxygen conditions, in both conditions, 3 or 20% oxygen, a significant difference was found in the simple effect of time (*p* = 0.00, F = 95, η^2^ = 0.85, *p* = 0.00, F = 121, η^2^ = 0.88).

### 2.5. Principal Component Analysis

In Figure 3, the integrated parameters assess and reduce the number of variables to identify the most relevant outcomes and explain the effects found in this study. Analysis showed that PC1 in the biplot and PC2 explain 52.7% of the total variance. It suggests that both components capture more than half of the information contained in the original variables. Furthermore, variables are represented as vectors and the directions and lengths indicate their contribution to the components. PC1 separates the groups at T0 from the groups in T1 (Active-T0, Sham-T0 and Active-T1, Sham-T1) and explains 35.9% of the variance in the data. The most significant variables explaining the variability in PC1 were the microglia proliferation and viability (PRO3, PRO20, CV20, CV3) and the BDNF quantification. PC2 tends to separate the groups that received the active condition from the sham (Active-T0, Active-T1 and Sham-T0, Sham-T1). This component explains the 16.78% variance and the variables that contributed more were SCIP.IL, SCIP.DL, and SCIP.VF (Appendix A).

## 3. Discussion

In the first stage of this randomized, double-blind, sham-controlled clinical trial, a 3-week intervention was conducted in which patients received 15 sessions of active tDCS or sham tDCS plus 9 sessions of CS. The results suggest that significant effects of CS on overall cognitive performance and disease severity were observed after the intervention. Likewise, notable results were also found in the quantification of IL-6, CX3CL1, and BDNF, though no differences were observed between groups. Similarly, significant results were found in the proliferation and survival of microglia cultured under standard oxygen tension conditions (20%) but treated with patient aMCI serum. Despite the combination of tDCS and CS, both groups benefited equally, which may be attributed to the effects of the CS. Interestingly, differences favoring the real group were observed between the groups when microglia was cultured under 3% oxygen tension conditions in the presence of aMCI serum.

### 3.1. Cognitive Stimulation Modifies Cognition in aMCI

Regardless of group assignment, patients showed similar scores in clinical and demographic data before starting the study, reflecting the homogeneity between groups. In this study, we provide evidence that the combination of CS with tDCS was not superior to CS with sham tDCS in terms of global cognitive performance, as assessed by the total MoCA, and disease severity, measured by the CGI-S, compared to baseline. These findings are consistent with those reported in four controlled trials conducted in patients with MCI [35,36,37,38], which yielded similar results. In contrast, two other studies, including the largest to date by Hanna Lu (2019) with 201 participants, and the study by Rodella (2022), also evaluated the combination of computerized cognitive training with tDCS [31,32]. However, neither of these studies employed an in-person group modality, unlike Gu (2022) [53], who administered only five sessions of tDCS without CS and found changes in memory, and also Stonsaovapak (2020) [54], who provided 10 sessions of tDCS without CS and reported changes in visual sustained attention, visual memory, and working memory. Yet neither study reported improvements in global cognitive performance. In addition, one systematic review and another systematic review with meta-analysis found that active tDCS improves overall cognitive function compared to sham tDCS at the end of 10 sessions, but with no benefits. A key consideration in both reviews is that they did not differentiate between interventions that directly influence cognition, such as cognitive stimulation (CS), and others like physical exercise, which may have confounded results [33,34]. Thus, cautious interpretation and more extensive trials with an increased number of sessions are suggested. Therefore, testing the additive effect of tDCS and group-based CS remains an open question. Also, given the heterogeneity among studies, assessing the efficacy of tDCS in improving cognitive function in individuals with MCI is challenging, particularly in amnestic subtypes, considering that it may represent a progressive disease. This variation may be attributed to the need for specific instruments to measure particular cognitive domains in order to detect differences. Thus, establishing a standardized protocol is one of the main difficulties [2,7]. Although no significant cognitive differences were detected between groups in our study, the results are relevant for identifying optimal parameters that could enhance the efficacy of neuromodulatory interventions combined with CS. Patients reported perceived benefits, which were quantified using the MFEQ. This self-report questionnaire is designed to assess the frequency and impact of memory failures in the daily life of a person. As a self-report instrument, it relies on the subjective perceptions of individuals regarding their cognitive functioning and the difficulties they may encounter in their daily life [55]. While the MFEQ indicates a trend in the results of this study, it aligns with the fact that patients perceived benefits; most noted an implicit improvement in their experience, which was documented during clinical assessment. Furthermore, both interventions were well tolerated, with no adverse effects reported, confirming their safety in clinical practice.

Although a systematic review indicated that tDCS could be considered a therapeutic proposal for aMCI and AD by improving global cognitive function, there are other NIBS techniques, such as repetitive transcranial magnetic stimulation (rTMS), that are more effective in these patients [19]. Unfortunately, we have not observed an apparent effect of tDCS in the first-stage results. A possible explanation for these differences could be that the response to the intervention follows a pattern similar to the progression of MCI, where neurobiological phenomena emerge prior, followed by clinical symptoms [56]. However, it is necessary to note that 15 tDCS sessions may not be enough to appreciate the changes at the cognitive and plasticity level, so finding the optimal stimulation parameters regarding the number of sessions of tDCS is a challenge in neuromodulation since these can vary according to the integrity of the brain to be stimulated [57]. In this regard, other studies using tDCS to address AD or MCI faced difficulties in establishing a measure of change, as the number of sessions is concentrated in too short a period for adequate reassessment [58]. Nevertheless, additional research supported the beneficial effect of a low number of tDCS sessions in the presence of computational cognitive training [59]. As a result, the instruments used fail to capture subtle changes or provide longitudinal patient follow-up in areas affected in the earliest stages of the disease, such as memory and learning. Later, other domains are compromised, such as executive functions, language, and visuospatial and visuoconstructive skills [60,61]. Additionally, measuring cognitive symptoms in MCI presents challenges due to the subtle nature of the deficits. The assessment scales used may lack the necessary sensitivity to detect small changes in cognitive functioning in individuals with aMCI [62]. Some research employs unsuitable instruments, such as the Alzheimer Disease Assessment Scale-Cognitive (ADAS-COG) and the Mini-Mental State Examination (MMSE), or measurement parameters that do not reflect early features of cognitive decline [29,30,31,58]. Other studies have encountered similar challenges with these instruments, failing to demonstrate significant changes; however, their results indicate trends toward cognitive improvement [36]. Thus, we also considered that, in aMCI, the impression of the patients of improvement is crucial, and when assessed with MFEQ, this also resulted in a positive trend. In this regard, a previous study proposed that it is important to take into account that statistical significance does not represent the benefits experienced in their entirety in each patient, i.e., clinical significance, so it is recommended to verify the impact of the intervention on the daily lives of patients in future studies [63].

Studies that have reported null results with tDCS in their initial hypotheses have nonetheless provided findings that justify further research into the combination of tDCS and cognitive training. For instance, González (2021) reported that this combination improved processing speed in specific tasks more than sham tDCS or cognitive training alone [36]. Simko (2024) observed alterations in right insular activity through fMRI, suggesting that multimodal approaches may help better understand how intensified tDCS protocols affect brain plasticity and cognitive performance [37]. Antonenko (2024) found an increase in the functional connectivity of the frontoparietal network, which could indicate a long-term modulation of neural networks, although its clinical relevance remains uncertain [38]. Moreover, the relationship between individual improvements and the magnitude of the electric field points to a possible dose–response effect. Finally, Martin (2019) observed unexpected improvements in verbal memory at a three-month follow-up, reinforcing the need for continued exploration of this combined intervention [35]. Taken together, these studies suggest that while the effects of tDCS combined with cognitive training may not be immediate, the approach has the potential to generate long-term benefits that warrant further investigation. However, studies in MCI have shown heterogeneous results, partly due to variability in experimental designs. While tDCS may induce synaptic plasticity and enhance neuronal viability through BDNF production [64], it has not yet been demonstrated to significantly slow progression to dementia. In our study, no cognitive or clinical differences were observed when comparing cognitive training in the presence of sham tDCS. Research in this domain remains active, encountering various methodological challenges, yet there are high expectations that repeated sessions combined with group cognitive training could amplify long-term effects.

### 3.2. IL-6 Concentrations Indicate the Benefits of Cognitive Stimulation Effects in aMCI

Several studies have shown that elevated levels of IL-6 increase the risk of cognitive and memory decline [65], even in individuals without a dementia diagnosis [66]. In this context, our study shows that CS has a differential effect on the proinflammatory cytokine IL-6 and the BDNF in the serum of patients with MCI. In this sense, altered concentrations of proinflammatory cytokines in the periphery have been strongly linked to elevated IL-6 levels in the blood, which are associated with reduced gray matter volume in the hippocampus [67,68], decreased total brain volumes [69], and a greater rate of cortical thinning over time [70] analyzed with magnetic resonance imaging. Also, it is known that IL-6 is closely related to learning and memory disorders [67] and can contribute to cognitive dysfunction by activating various cellular signaling pathways and interfering with synaptic plasticity and long-term potentiation [68]. In this context, the downregulation of IL-6, mediated by CS, could improve cognition in patients with aMCI. However, the mechanisms underlying this effect have not been described in humans, although animal models exposed to cognitive stimulation through exposure to an enriched environment show a decreased production of proinflammatory cytokines such as IL1β, IL-6, and TNF-α through inhibition of NLRP3 inflammasome activation and autophagy [71].Additionally, dysregulation of IL-6 in the bloodstream could contribute to cognitive impairment via synaptic regression and apoptosis of cells, dysregulation of neurotransmitters, demyelination, or neuronal death [71]. Regarding to the benefits of cognitive stimulation through enriched environment, the regulation of inflammatory signaling could occur through the reduction in the expression of tumor necrosis factor receptor (TNFR)-associated factor 6 (TRAF6), a key binding protein in the inflammatory signaling pathway activated by ascending inflammatory signals [69]. TRAF6 can interact with IRAK1, promote nuclear localization, and transactivation activity of NF-κB p65, which regulates proinflammatory cytokines such as IL-6 [70]. Taking this into account, it is possible that the reduction in IL-6 observed after intervention with CS and tDCS could help mitigate this decline and improve cognitive functions, potentially counteracting the deleterious effects of IL-6 on brain health. However, it is important to note that animal models also suggest that cognitive stimulation, mimicking enriched environments, increases BDNF levels, which play a key crucial role in cell survival and neuronal maturation [65,72]. This neurotrophin acts through the tropomyosin receptor kinase B to activate several signaling pathways involved in the modifications of neuroplasticity [73,74,75].

The above-mentioned studies may support the relevance of our findings. An elevated concentration of IL-6 and a low concentration of BDNF were observed in these patients. However, after the interventions, an opposite effect on IL-6 and BDNF concentrations was observed, both in the presence and absence of active tDCS. Similarly, the concentration of CX3CL1 also showed changes following stimulation. Thus, the IL-6 and BDNF concentrations may be in the direction of the cognitive improvements seen in the aMCI patients of our study.

### 3.3. Human Microglia Improved Proliferation and Viability with Serum of aMCI After the Intervention with Cognitive Stimulation

In addition to the modifications found in peripheral factors, it is known that inflammation in the brain involves the participation of different molecules, such as IL-6, that could regulate changes in microglia; these changes could occur even before cognitive symptoms appear [76]. Also, several studies showed bidirectional communication between the brain and the peripheral immune system [77,78,79]. For instance, in aging, an inflammatory state is present, and this could lead to the disruption of the blood–brain barrier and promote cognitive impairment through the peripheral inflammatory cytokines such as IL-6 that participate in the regulation of BBB [80,81,82,83,84]. Proinflammatory cytokines from the periphery, such as IL-6 or CX3CL1, could modify the response of microglia, increasing the proinflammatory environment and cognitive impairment [85].

Microglia, the mononuclear phagocytic cells of the central nervous system, play essential roles in health and disease. They are the first line of defense in the central nervous system against infection and injury [86]. They also contribute to neurogenesis [87], participate in synaptic pruning [88], and support neurons and other glial cells [89]. Their essential functions and protective roles reassure us that the body has robust natural defense mechanisms against neurological pathologies, including the age-related neurodegenerative condition of MCI, which can progress to Alzheimer [90].

Microglial activation has been extensively studied in MCI. However, these studies have been limited to working with patients using neuroimaging techniques such as Positron Emission Tomography [91,92]. This study focused on whether soluble factors in patient serum regulate microglial proliferation and survival. Thus, our results clearly showed a decrease in microglial proliferation and survival, which was associated with high levels of IL-6 in the serum of patients before the CS with or without tDCS. These observations are consistent with other studies indicating that chronic exposure to IL-6 can overstimulate microglial cells, leading to a desensitized phenotype or even apoptosis [93,94].

Furthermore, IL-6 overexpression has been implicated in the potentiation of apoptosis in vitro models through activation of the Fas-mediated extrinsic apoptotic pathway [95]. However, the present study suggests a potential solution. After CS with or without tDCS, both microglial proliferation and viability increased to values close to those of the control group. This promising result suggests that CS has the potential to modulate the decrease in inflammatory cytokines like IL-6, which regulate microglial proliferation and survival. This potential breakthrough is significant as microglial cells have been shown to play a protective role during the early stages of MCI [96].

This research, which involves using two conditions of oxygen tension for microglial cell culture to mimic the physiological oxygen tension 3% (physiological oxygen tension), rather than the standard conditions of oxygen tension (20%) generally used for cell culture, has the potential to inspire curiosity and interest. In both culture conditions, we found simple effects; however, when microglia were cultured with 3% oxygen in the presence of the serum samples, a significant influence of the tDCS and CS combination was detected [97]. This suggests that, at least in this parameter that may be related to the inflammation, the combination of both interventions promotes higher effects on microglia proliferation. This finding opens the possibility of exploring the relevant role of soluble factors in the serum of aMCI patients on microglia as a surrogate cellular model to show the effects of CS in the presence or absence of the tDCS. Thus, the observation seen in microglia points in the same direction as the benefits of tDCS in influencing the excitability of cortical neuronal plasticity and regulating the expression of proteins that promote neuronal differentiation and survival, such as BDNF [64]. A study with older adults with MCI who received CS intervention showed higher BDNF levels and improved cognition. Thus, the mechanism underlying this effect is not known [98]. In support of the observations presented in our study on microglia, the preliminary results of IL-6, BDNF, and even CX3CL1 or the proliferation and viability of microglia cells strongly suggest that maintaining the tDCS after the initial priming carried out with its combination with CS may show benefits (data not published). However, this observation needs to be confirmed with more participants. Additionally, the benefits of tDCS in combination with CS require further investigation in additional studies.

### 3.4. Effects of CS and tDCS on Cognitive, Biological, and Microglial Variables: A PCA Approach

According to the results of the PCA, the first principal component PC1, suggests that CS induces memory-related changes that are not enhanced using of tDCS when the MOCA-Mis is modified. It suggests that patients fail to evoke, but the intervention with CS helps in the memory subprocesses of storage and encoding, but not in evocation since they continue to depend on external cues, whether semantic or recognition. It also seems that other variables such as PRO3, PRO20, CV20, CV3, BDNF, IL6 and CGI-S are highly involved when patients undergo this type of intervention, interestingly, it is the clinical impression the only one that is not molecular and before we mentioned how the patients and the clinician perceived a positive modification that is not fully evident in the clinimetry instruments. The PC2, suggests that the use of tDCS could explain certain differences that are present in the groups that received the combination of CS plus tDCS. Curiously, total MoCA is a difficult measure to modify in aMCI, and the fact that its score does not increase and remains the same is considered a positive result. This is in line with other trials carried out with medications [7], in addition to the fact that PC2, although to a lesser extent, manages to show other cognitive variables, the SCIPS-IL, which represents a measure of encoding; then SCIPS-DL, which represents evocation; and, the SCIPS-VF represents the ability to evoke elements of a given category and SCIPS-Total, which is a global measure of cognitive functioning. The PCA revealed a clear differentiation between the variables in the CS plus tDCS active and CS plus tDCS sham groups for the group by time interactions.

### 3.5. Strengths

One of the strengths of this study is addressing brain mechanisms to improve or maintain cognitive reserve while reducing inflammation, and increase social contact through interventions that allow addressing some of the potentially modifiable risk factors in dementia to slow down the neurodegenerative process [10,13]. Another strength is that we focused on using interventions supported by the evidence available so far. First, we stimulated the l-DLPFC, as it is considered a key neuroanatomical substrate for the neuronal adaptation of cognitive functions. Furthermore, its accessibility and connectivity with subcortical structures involved in MCI suggest that it might be sufficient to focus solely on this area [21,99]. This could enhance processing efficiency by directly modulating the neural network to correct pathological patterns of brain activity and induce more functional activity [100]. Previous studies have emphasized the role of l-DLPFC in cognitive functions such as verbal memory, working memory, and executive function. The effectiveness of stimulation in this area depends on factors such as session number, intensity, and other parameters. Based on these findings, we consider that stimulation on the l-DLPFC could be sufficient [33,99,101,102]. Studies that directly compare the two regions could further strengthen this hypothesis. Furthermore, we carefully selected stimulation parameters, such as an intensity of 2 mA [103], a session duration of 30 min and the application of multiple sessions which are recommended to maximize effectiveness [37], as post-effects do not always follow a linear pattern. Some evidence has shown that prolonged session duration may alter excitability, and therefore regular and frequent repetition of sessions is suggested to improve outcomes [104].

tDCS was applied offline to the CS, as it has been shown in healthy individuals to be the most effective learning facilitator option for working memory tasks compared to online or placebo [40,41]. On the other hand, there are different CS protocols, which can be individual or personalized [36,105]. We present a group approach to promoting social contact, a relevant modifiable factor for healthy aging, and emphasize that this is one of the protective elements that can help slow the progression from MCI to dementia [10,13]. It should be noted that the number of CS sessions was not equal to the number of tDCS sessions. This decision followed the Memory Unit of Madrid Salud recommendation that more than three CS sessions per week may be stressful [106]. To address this, it is suggested that the level of difficulty be gradually increased as the intervention progresses [106]. Finally, the study design allowed for the verification of the effects of CS with biomarkers, which distinguishes it from studies that evaluate it only with psychometric test, we seek to support our results with molecular biomarkers, considering that this is a neurodegenerative disease with no cure to date, which underscores the need to continue exploring alternatives based on the results of the component analysis, which will allow us to select variables that truly manage to explain the results. To the best of our knowledge, this is the first study to use serum biomarkers to assess the state before and after intervention with CS and tDCS in MCI, which can help to explain the mechanisms underlying cognitive changes and guide future interventions.

### 3.6. Limitations

One limitation of this study is the use of a single version of the MoCA, as no validated alternate versions were available at the time of evaluation. Although the possibility of a learning effect is acknowledged, its impact was likely minimal given the 3-week interval between assessments [107] and considering all participants evaluated were cognitively impaired [108,109]. Likewise, maintaining the same version of the instrument guarantees that any modification in the score is attributable to changes in the patient’s cognition and not to differences between the items implemented, which may favor reliability and consistency [110]. Since both the CS plus tDCS active and CS plus tDCS sham groups underwent identical tests, any differences in improvement can be attributed to the intervention, assuming the learning effect affected both groups equally. To overcome this, we used the SCIP-S, which is a test that allows for the assessment of different cognitive domains using two of the three different parallel forms. However, this does not solve the problem entirely, since the normative interpretation of the SCIP-S is carried out from transformed scores expressed in percentiles and T scores, while in this study we chose to use natural scores since it is not adapted to the Mexican population [111]. To further address potential biases, mixed models were used in the statistical analysis to control baseline variations and isolate the effect of the intervention.

Also, our study does not allow us to distinguish the effects of tDCS from CS, as we did not include a tDCS-only group or a sham tDCS group with sham cognitive training. However, it is important to mention that we designed the study to assess the impact of increasing session repetition and frequency—key factors in tDCS effectiveness according to previous studies on plasticity and cognition [37,112]. Additionally, group cognitive training has been shown to improve adherence and motivation while facilitating the transfer of benefits to daily life [20].

The decision not to include a tDCS-only group or a sham tDCS group with sham CS was driven by our interest in specifically evaluating the combination of tDCS with group training under a high-repetition and high-frequency scheme. Furthermore, this strategy optimized study feasibility, as no participants have dropped out to date. To the best of our knowledge, no prior studies have reported a protocol with these characteristics. Our study provides novel evidence on the feasibility and benefits of this approach, which could guide future research and clinical applications for treating cognitive deficits [113]. In this study, there was a higher proportion of women (77.78%) than men (22.2%). While this disparity could introduce potential bias in the findings, as there may be notable differences in responses between genders, exploring such differences was not within the scope of our objectives. In the same way, the sample size seems to be a limitation. A power of 70% is required to modify the results obtained with the current data. Even if the sample size is increased, it is unlikely to detect a significant effect with tDCS in aMCI after 15 sessions, suggesting that the hypothesis is null. However, we believe that our measures of change, both soluble and microglia factors, could be useful in future studies with a larger number of sessions and longer duration. We opted for 15 sessions of tDCS to optimize the response, increasing the number of sessions compared to previous studies [112]. This decision was based on our prior study using rTMS in mild to moderate AD, in which we applied 15 sessions over three weeks, comparing two modalities: one with exclusive stimulation of the l-DLPFC and another targeting six areas relevant to AD, without cognitive training [101]. Both approaches equally improved cognition after three weeks of treatment, and this effect persisted for four weeks without further intervention. However, we acknowledge that the optimal duration of the intervention remains a limitation of our study. Future research should consider extending the duration of tDCS interventions to explore these potential differences more comprehensively.

So far, this does not mean that tDCS lacks cognitive or plasticity effect, but that further studies are needed to determine the optimal stimulant dose to treat aMCI, as suggested by research looking into possible hermetic mechanisms of tDCS when applied at negligibly low current intensities [114]. These studies should consider the intensity, number and duration of sessions, as well as application sites, continuing to assess the molecular components that mediate the response prior to cognitive and clinical changes, which could be useful as a biomarker of response to intervention. Here, it may be discouraging that the combination of CS with tDCS does not show an effect in MCI, as was observed with NIBS in other diseases. For example, in depression, the combination of tDCS plus Selective Serotonin Reuptake Inhibitors accelerates the response [115,116]. However, given that in MCI available interventions are limited, this could be encouraging, as even minimal changes are considered clinically important for outcomes and could slow the progression to dementia, although this assertion needs to be probed in other studies [117].

## 4. Materials and Methods

### 4.1. Trial Design

A randomized clinical trial (RCT) is being conducted, which is double-blind and sham-controlled. This study focuses on the partial results obtained after 3 weeks from two groups: (1) active tDCS plus CS and (2) sham tDCS plus CS with two measurements per participant, T0 corresponding to the baseline measurements and T1, after applying 15 continuous sessions of tDCS, both active or sham, and at the same time, 3 CS sessions per week, 9 in total. Figure 4 shows the study design in detail that was conducted at the Clinical Research Division of the Ramón de la Fuente Muñiz National Institute of Psychiatry in Mexico City, Mexico, and all procedures were approved by the Institutional Ethics Research Committee and registered (CEI/P/017/2020; ClinicalTrials.gov NCT06467253). Further details, including sample size calculations, are included in the Appendix A, Study Recruitment. All participants involved in the study were informed and provided written informed consent in accordance with the Declaration of Helsinki of 1975. Participants received no financial compensation, and the study was free of charge. This study followed the Consolidated Standards of Reporting Trials (CONSORT) reporting guidelines. Appendix A shows the CONSORT Flow Diagram with study enrolment, visits, and attrition.

### 4.2. Randomized and Blinded

Participants were randomly assigned to receive one of two interventions. All participants first received active tDCS or sham tDCS and subsequently received CS in groups of up to four patients to receive the benefits of group CS. Double blinding was achieved by using sham sessions with equal instrumentation, sensation, and duration. For sham-tDCS, the technician was provided with preprogrammed equipment under the same physical and handling conditions as the active equipment. Random assignment of tDCS codes was performed by an independent investigator who was not involved in treatment nor assessments. The randomization sequence was generated using the website www.randomization.com, accessed on 13 June 2022. Patients, assessors and technicians remain blinded to the study conditions. Only data analysts had access to that information.

### 4.3. Patients

The study included women and men aged 60 to 75 years with suspected aMCI who were recruited through in-clinic outreach at the Instituto Nacional de Psiquiatría “Ramón de la Fuente Muñiz” in Mexico City and the Geriatric Clinic of Coyoacán, as well as through social media advertisements. All participants met the modified Petersen diagnostic criteria [1,118]: 1. Concern about a change in cognition reported by the patient, an informant, or a trained clinician. 2. Impairment in one or more cognitive domains relative to the individual’s age and educational level. A total score on the MoCA between 19 and 25 was considered for inclusion in the study. To confirm the diagnosis, neuropsychological tests were conducted, revealing below-average performance. Specifically, performance had to be between 1 and 1.5 standard deviations below the mean, adjusted for age, education, and cultural background. 3. Preservation of global function, with independence in functional abilities, although tasks may require more time or be performed with reduced efficiency. 4. Exclusion of other causes: the diagnosis of MCI required ruling out the presence of an alternative medical, psychiatric, or neurological cause that could explain the observed deficits. For this purpose, biochemical studies were conducted. Data were collected regarding educational level, employment, as well as information related to modifiable risk factors [10,13] and comorbidities, which were not used as variables due to heterogeneity. In case of psychiatric conditions such as depression, patients were first required to receive treatment for three months to avoid diagnostic bias. A careful screening was performed to exclude individual with a history of neurological or brain surgery, stroke, brain tumor, aneurysm, severe head injury or other significant neurological disease, as well as those with uncontrolled diabetes mellitus, hypertension, dyslipidemia, infections, thyroid disorders, or vitamin deficiency. All participants received an initial diagnosis upon entry into the study Appendix A. The full list of inclusion and exclusion criteria is provided in Appendix A.

### 4.4. Cognitive Stimulation Program

A CS program designed by the Memory Unit of Madrid Salud for aMCI was adapted into Mexican Spanish [106]. This program aims to stimulate various cognitive domains including memory, attention, perception, concentration, temporal orientation, constructive praxis, processing speed, language, daily activities, calculation, and planning. It also includes specific habits, strategies, and techniques such as visualization, association, memory of texts, memory of names, and everyday forgetfulness. The CS consisted of 9 sessions, each lasting 90 min. The sessions were conducted in groups with a maximum of 4 participants, three times a week—as stipulated by the Madrid Salud Cognitive Impairment Prevention Center—gradually increasing in complexity and difficulty to avoid frustration and discomfort for the participants [106]; following a tDCS session (active or sham). The sessions began with a guided relaxation exercise, followed by an introduction to the domain or strategy to be reviewed and a reflection on its application to daily life. Oral exercises on calculation, language, and remembering words and texts were then conducted. Participants were also given explanations about the activities to be performed at home. Then, temporal orientation was reinforced by reminding them of the day, month, year, and day of the week so they could finally start the resolution of pencil-and-paper exercises. At the end of the 9 sessions, a group reflection was held on applying and maintaining the new habits and strategies in daily life (Appendix A).

### 4.5. Transcranial Direct Current Stimulation

tDCS procedures complied with the agreed guidelines for safe tDCS delivery endorsed by the International Federation of Clinical Neurophysiology [115,119]. Stimulation was delivered offline prior to CS. tDCS was delivered using portable devices that are approved for medical use (Sooma tDCS^TM^ version A12, Sooma Oy, Helsinki, Finland) and run on 9v batteries. They have the option of active or sham configuration. Active tDCS sessions consisted of a current input phase that was gradually adjusted to an intensity of 2 mA for a duration of 30 min. Sham tDCS sessions had the same duration, but the current of 2 mA decreased to zero as the first 30 s of stimulation progressed. The setup was guided with a cap system used by sizes for the standardized placement of two 25 cm^2^ (5 × 5) electrodes with the anode placed over l-DLPFC (F3) and the cathode over r-DLPFC (F4) according to the 10–20 electroencephalography system. The effectiveness of the stimulation was monitored by the equipment, which includes safety features that measure the contact between the electrode and the scalp, interrupting the stimulation if poor contact is detected. This was also continually checked by the technician overseeing the procedure.

### 4.6. Enzyme-Linked Immunosorbent Assay

BDNF, CX3CL1, and IL-6 were quantified in the serum of aMCI patients at T0 and T1 using enzyme-linked immunosorbent assay (ELISA) kits according to the protocol provided by the manufacturer. Commercial ELISA kits were used to assess serum BDNF and CX3CL1 concentrations (R&D Systems, Minneapolis, MN, USA). For the assessment of serum IL-6 concentrations, a human high-sensitivity ELISA kit was used (Enzo Life Science, Inc., Farmingdale, NY, USA). Plates were read with a Glomax Discover microplate reader (Promega, Fitchburg, WI, USA).

### 4.7. Cell Culture and Treatment with Serum

Human HMC3 microglial cells were purchased by the American Type Culture Collection (ATCC-CRL3304, Manassas, VA, USA). Cells were cultured in Dulbecco Modified Eagle Medium F12 (DMEM/F-12) supplemented with 1% penicillin/streptomycin and 10% fetal bovine serum (FBS) (Thermo Fisher Scientific, Waltham, MA, USA) and maintained in a humidified incubators at 37 °C (Binder, Bohemia, NY, USA). HMC3 cells were cultured in two different oxygen conditions: 3 and 20% until 80% confluency. Thus, cells from four different passages (1 to 4) were plated in 96-well plates at a density of 3 × 104 cells per well and cultured for 24 h in 10% FBS to allow their attachment. Then, HMC3 were deprived of FBS for 24 h [69] to add serum from aMCI patients. Serum was filtered with 0.22 μm membrane previously to add to the cultures. Serum samples corresponded to the initial time (T0 or basal) and to the end time (T1 or post intervention). Positive control with 10% FBS was included as a reference (Figure 2B).

### 4.8. Cell Viability and Proliferation Assay

Cell viability was determined at T0 and T1 using a Cell Proliferation Reagent WST-1 (Sigma-Aldrich, St. Louis, MO, USA) by following the instructions provided by the manufacturer. The proliferation assay was performed by quantitative determination of BrdU incorporated into cellular DNA using the 5-Bromo-2′-deoxy-uridine Labeling and Detection Kit III (Sigma-Aldrich, St. Louis, MO, USA) following the manufacturer instructions. Cell absorbance at 405 nm for the viability assay and at 450 nm for the proliferation assay was measured with a Glomax Discover microplate reader (Promega, Fitchburg, WI, USA).

### 4.9. Cognitive and Clinical Outcomes

Clinical evaluations of the effect of interventions were measured at T0 and T1 with the following instruments: score MoCA Total Score (MoCA Total); MoCA Memory Domain Score (MoCA Mem) MoCA Memory Index Score (MoCA MIS); Memory Failures of Everyday Questionnaire (MFEQ); Clinical Global Impression-Severity (CGI-S); Clinical Global Impression-Improvement (CGI-I); Screening for Cognitive Impairment in Psychiatric (SCIP-S); SCIP-S Immediate Verbal Learning Score (SCIP-S IV); SCIP-S Working Memory Score (SCIP-S WM); SCIP-S Verbal Fluency Score (SCIP-S VF); SCIP-S Delayed Verbal Learning Score (SCIP-S DVL); SCIP-S Speed of Processing Score (SCIP-S SP) and SCIP-S Total Score (SCIP-S Total) Appendix A.

### 4.10. Statistical Analysis

Data analysis was carried out using the program R, version 2023.06.2+561 [120]. Demographic and clinical data of patients were analyzed using descriptive statistics. The Shapiro–Wilk normality test (*p* < 0.05) was performed. It was decided to use non-parametric statistics since the assumptions of normality and homoscedasticity were not met. Clinical, cognitive, cortical excitability, biological and microglial response variables neither follow a normal distribution, so ARTool in RStudio was used, which performs an aligned rank transformation, enabling the execution of a nonparametric factorial ANOVA in models with fixed and random effects (or repeated measures) for each response variable [121,122]. The fixed factors were group (active, sham) and time (T0, T1), each with two levels. A grouping term (ID) was included since there were two response measurements per patient. This term in ARTool determined a mixed-effects linear model. In this way, the effect of each factor (group and time) was assessed and at the same time their interaction, based on the two measurements taken for each subject. Finally, the effect size was calculated with partial eta squared (η^2^). Additionally, we carried out a principal component analysis (PCA) to identify new variables, the principal components, which are linear combinations of the original variables that explain the greatest proportion of the variance in the group × time interactions [123] Appendix A.

## 5. Conclusions

Based on the results of this study involving the combination of CS and tDCS, it is suggested that after 15 sessions of tDCS and 9 offline sessions of CS, the cognitive effect appeared to be solely attributable to CS. Although the initial hypothesis was not confirmed, the findings provide valuable insights. However, a potential influence of tDCS combined with CS was observed, which may be related to inflammation and could promote microglial proliferation. It appears that tDCS makes a positive contribution to the proliferation of microglial cells cultured with 3% oxygen tension, suggesting that the immediate effects of the combined intervention may be at the biological level and may not yet be evident at the cognitive level, as is suggested with the PC1 component of the PCA. The PC1 component is primarily associated with molecular variables PRO3, PRO20, CV20, CV3, BDNF, IL-6, and one clinical variable, CGI-S. It successfully differentiates the evaluation time in both groups.

Moreover, the findings of our study open up several avenues for future research. First, it is necessary to assess whether a longer duration of tDCS could produce more pronounced effects on cognitive function and brain plasticity, justifying its combination with group CS. Second, the observed molecular changes could serve as biomarkers of response to the intervention, requiring further studies to validate their clinical utility. Finally, the optimal number of sessions and stimulation interval have not yet been determined, so future research should explore these parameters to optimize the effectiveness of tDCS in populations at high risk of progressing to dementia, such as aMCI. Thus, tDCS at 2 mA over the l-DLPFC may be a promising intervention for aMCI, given the risk of progression to dementia, a condition that remains incurable and whose prevalence is inexorably increasing with global aging. In summary, tDCS did not enhance the cognitive or modulatory effects of CS, suggesting that longer interventions might be required to achieve substantial benefits. Although tDCS has shown potential to improve cognitive function in MCI, its impact on progression to dementia has yet to be established. Further studies are required to determine its efficacy and safety in this context.

## Figures and Tables

**Figure 1 ijms-26-01754-f001:**
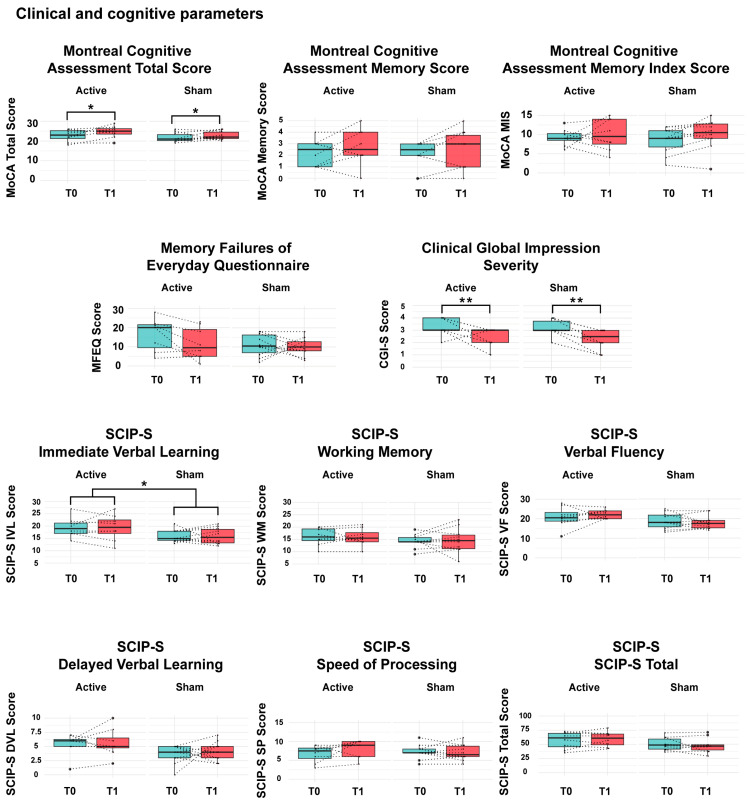
Clinical and Cognitive Parameters. Includes Montreal Cognitive Assessment Total (MoCA Total), Montreal Cognitive Assessment Memory Domain Score (MoCA Mem) and Montreal Cognitive Assessment Memory Index Score (MoCA MIS), Memory Failures of Everyday Questionnaire (MFEQ) and Clinical Global Impression-Severity (CGI-S) scores. Screening for Cognitive Impairment in Psychiatry (SCIP-S) includes Immediate Verbal Learning, Working Memory, Verbal Fluency, Delayed Verbal Learning, Speed of Processing and Total SCIP-S. ** *p* = 0.01; * *p* = 0.05.

**Figure 2 ijms-26-01754-f002:**
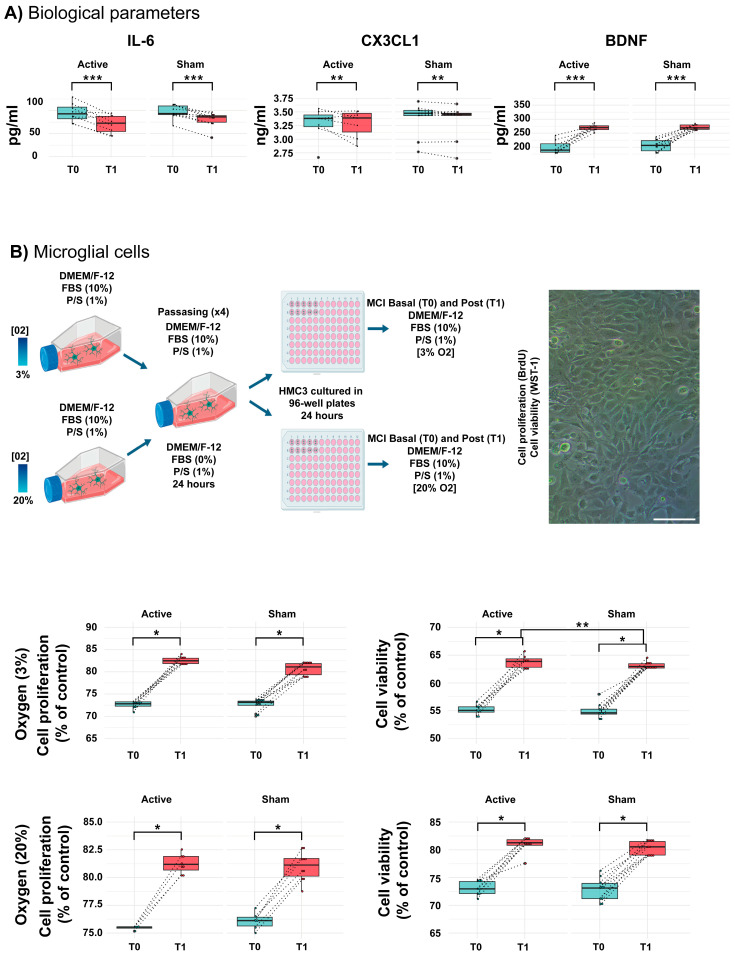
Biological parameters. (**A**) ELISA for interleukin-6 (IL-6), CX3CL1, and BDNF from the serum of patients with aMCI treated with active tDCS plus CS or sham tDCS plus CS. Protein quantifications (picograms/milliliter (pg/mL) and nanograms/mL (ng/mL)) were performed before (T0) and after (T1) the application of the interventions. Results are represented as the mean + error standard of the mean (SEM). (**B**) Experimental design for microglia cell culture. Microglia was cultured on 3 and 20% of O2 in the presence of fetal bovine serum (FBS) and penicillin/streptomycin (P/S). Thus, cells were plated on 96-well plates to be treated with serum of the participants with aMCI to measure cell proliferation and viability. A micrography of microglial cells is shown on panel (**B**). Scale bar = 30 μm. Charts show proliferation (left side) and viability (right side) of microglia cells cultured in 3 or 20% oxygen, respectively. *** *p* = 0.001; ** *p* = 0.01; * *p* = 0.05.

**Figure 3 ijms-26-01754-f003:**
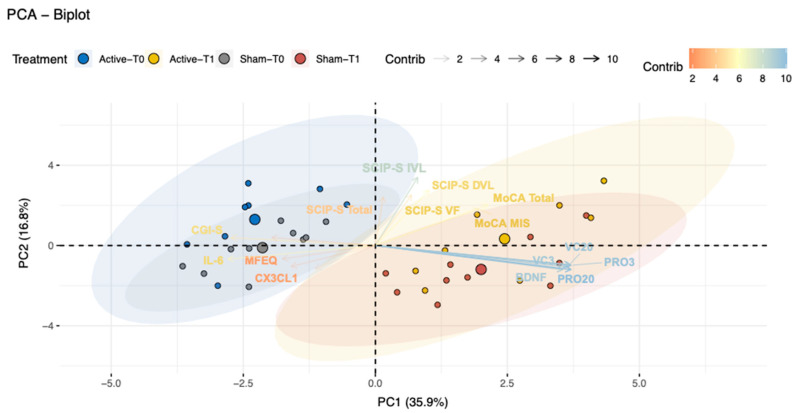
Principal Component Analysis (PCA). PCA was applied to identify patterns of variance between the combined intervention (active-sham) and the times evaluated (T0–T1). The color dots represent the category (active or sham) to which the variables belong. The ellipses show clustering patterns while the arrows indicate the strength and direction. The variables C20, PRO20, and BDNF appear to be positively correlated since their vectors are close and point in the same direction (to the right). Variables whose vectors point in opposite directions are negatively correlated, i.e., as PRO20 and BDNF values increase, IL6 and CX3CL1 (FKN) values tend to decrease.

**Figure 4 ijms-26-01754-f004:**
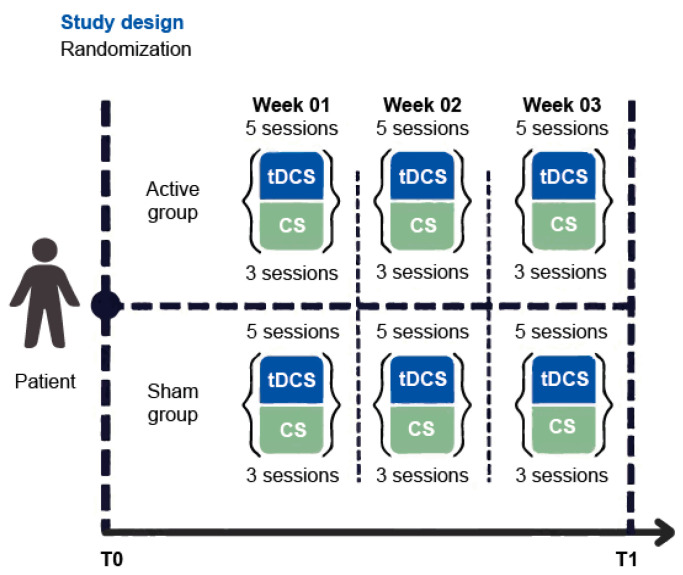
The randomized clinical trial design in amnestic mild cognitive impairment participants (aMCI) consists of assessing participants before (T0) and after (T1) the interventions. The interventions include the administration of 15 sessions of active (active group) or sham (sham group) transcranial direct current stimulation (tDCS), five per week. Together with 9 sessions of cognitive stimulation (CS), three per week. Both interventions were distributed over three weeks.

**Table 1 ijms-26-01754-t001:** Demographic and clinical characteristics.

	Groups	Sham (n = 10)	Active (n = 8)	Statistic Test
Age in years, mean ± SD ^1^		68.40 ± 4.74	68.250 ± 5.365	W ^14^ = 42, *p* = 0.8534
Years of education, mean ± SD		12.00 ± 4.03	13.750 ± 4.496	W = 44.5, *p* = 0.4242
Sex, n (%)	Male	2 (20%)	2 (25%)	*p* = 1
	Female	8 (80%)	6 (75%)	
Disease onset, n (%)				*p* = 1
	Subacute	2 (25%)	2 (33.3%)	
	Insidious	6 (75%)	4 (66.66%)	
Evolution disease, n (%)				
	Progressive	7 (77.77%)	7 (87.5%)	*p* = 1
	Fluctuating	2 (22.22%)	1 (12.5%)	
Time since onset (months), mean ± SD		21.750 ± 17.970	45.600 ± 32.199	W = 15.5, *p* = 0.2008
Clinical instrument, mean ± SD				
	HIS ^2^	1.000 ± 1.155	1.125 ± 1.246	W = 43, *p* = 0.8143
	CCI ^3^	2.700 ± 0.823	3.125 ± 0.641	W = 53, *p* = 0.2329
	Katz ADL Index ^4^	6.000 ± 0.00	5.938 ± 0.250	W = 35, *p* = 0.3143
	Lawton & Brody IADL ^5^	7.850 ± 0.366	7.938 ± 0.250	W = 47, *p* = 0.4232
	MFEQ ^6^	10.550 ± 5.135	15.375 ± 10.424	W = 55, *p* = 0.1976
	CRIq ^7^	318.500 ± 148.056	413.750± 134.698	W = 60.5, *p* = 0.0751
	GDS ^8^	6.850 ± 2.540	8.375 ± 3.160	W = 52.5, *p* = 0.2805
	NPI ^9^	5.650 ± 7.768	4.313 ± 5.400	W = 30.5, *p* = 0.4173
	CGI-S ^10^	2.750 ± 0.851	2.786 ± 0.802	W = 42, *p* = 0.881
	CGI-I ^11^	1.900 ± 0.876	2.429 ± 0.976	NA
Cognitive instrument, mean SD				
	MoCA Total Score ^12^	21.80 ± 2.347576	22.75 ± 3.058945	W = 49.5, *p* = 0.4193
	SCIP-S Total Score ^13^	50.20 ± 11.12355	57.25 ± 14.87808	W =50.5, *p* = 0.3735

^1^ SD, Standard Deviation; ^2^ HIS, Hachinski Ischemic Score; ^3^ CCI, Charlson Comorbidity Index; ^4^ Katz ADL Index, Katz Index of Independence in Activities of Daily Living; ^5^ Lawton & Brody IADL, Lawton & Brody Instrumental Activities of Daily Living Scale; ^6^ MFEQ, Memory Failures of Everyday Questionnaire; ^7^ CRIq, Cognitive Reserve Questionnaire; ^8^ GDS, Geriatric Depression Scale; ^9^ NPI, Neuropsychiatric Inventory; ^10^ CGI-S, Clinical Global Impression–Severity; ^11^ CGI-I, Clinical Global Impression–Improvement; ^12^ MoCA, Montreal Cognitive Assessment; ^13^ SCIP-S, Screen for Cognitive Impairment in Psychiatry; ^14^ W, Wilcoxon test.

**Table 2 ijms-26-01754-t002:** Comparisons of clinical and cognitive parameters.

	Simple Effect (Group)		Effect Size	Simple Effect (Time)		Effect Size	Interaction Effect (Group × Time)		Effect Size
	F	*p*	η^2^	F	*p*	η^2^	F	*p*	η^2^
**Clinical and Cognitive Parameters**									
MoCA Total ^1^	1.54312	0.232055	0.087961	4.80857	0.043445 *	0.231086	0.62579	0.440474	0.037640
MoCA Mem ^2^	0.17318	0.682826	0.0107081	3.47278	0.080845	0.1783403	0.14461	0.708739	0.0089573
MoCA MIS ^3^	0.00259665	0.95999	1.6226 × 10^−4^	1.98337814	0.17817	1.1029 × 10^−1^	0.00071544	0.97899	4.4713 × 10^−5^
MFEQ ^4^	1.3017	0.270686	0.075237	3.6250	0.075061	0.184714	2.4602	0.136323	0.133273
CGI-S ^5^	0.239974	0.6308740	0.0147767	14.265396	0.001651 **	0.4713435	0.071191	0.7930188	0.0044297
CGI-I ^6^	NA^13^	NA^13^	NA^13^	NA^13^	NA^13^	NA^13^	NA^13^	NA^13^	NA^13^
SCIP-S IVL ^7^	3.7535695	0.070554	1.9002 × 10^−1^	0.0012406	0.972339	7.7529 × 10^−5^	0.2356628	0.633937	1.4515 × 10^−2^
SCIP-S WM ^8^	0.806026	0.38261	0.0479605	0.041978	0.84025	0.0026168	0.010538	0.91951	0.0006582
SCIP-S VF ^9^	3.80391	0.068876	0.1920787	0.11944	0.734146	0.074096	1.25996	0.313869	0.278221
SCIP-S DVL ^10^	6.03853	0.025791 *	0.2739987	0.01698	0.897947	0.0010601	0.17544	0.680891	0.0108460
SCIP-S SP ^11^	0.23534	0.63417	0.014495	1.14222	0.30104	0.066632	1.74418	0.20519	0.098296
SCIP-S Total ^12^	2.56785	0.12861	0.1382953	0.05585	0.81618	0.0034785	0.75407	0.39803	0.0450079

^1^ MoCA Total, Montreal Cognitive Assessment Total Score; ^2^ MoCA Mem, Montreal Cognitive Assessment Memory Domain Score; ^3^ MoCA MIS, Montreal Cognitive Assessment Memory Index Score; ^4^ MFEQ, Memory Failures of Everyday Questionnaire; ^5^ CGI-S, Clinical Global Impression-Severity; ^6^ CGI-I, Clinical Global Impression-Improvement; ^7^ SCIP-S IVL, SCIP-S Immediate Verbal Learning Score; ^8^ SCIP-S WM, SCIP-S Working Memory Score; ^9^ SCIP-S VF, SCIP-S Verbal Fluency Score; ^10^ SCIP-S DVL, SCIP-S Delayed Verbal Learning Score; ^11^ SCIP-S SP, SCIP-S Speed of Processing Score; ^12^ SCIP-S Total, SCIP-S Total Score, ^13^ NA, not applicable. ** *p* = 0.01; * *p* = 0.05.

**Table 3 ijms-26-01754-t003:** Comparisons of biological and microglial parameters.

	Simple Effect (Group)		Effect Size	Simple Effect (Time)		Effect Size	Interaction Effect (Group × Time)		Effect Size
	F	*p*	η^2^	F	*p*	η^2^	F	*p*	η^2^
**Biological and microglial parameters**									
IL-6 ^1^	0.28719	0.59940	0.017633	29.20463	5.84 × 10^−5^ ***	0.646054	2.17818	0.15939	0.119824
CX3CL1 ^2^	0.1079	0.746808	0.0066986	7.0656	0.017182 *	0.3063263	3.7374	0.071103	0.1893556
BDNF ^3^	1.8227	0.19578	0.10227	96.4851	3.52 × 10^−8^ ***	0.85776	1.9141	0.18551	0.10685
PRO 3 ^4^	4.4825	0.050260	0.21885	104.9468	1.96 × 10^−8^ ***	0.86771	7.6335	0.013862 *	0.32299
PRO 20 ^5^	0.27872	0.60479	0.017122	115.97253	9.70 × 10^−9^ ***	0.878763	1.21902	0.28588	0.070795
CV 3 ^6^	1.39945	0.25410	0.0804307	95.20903	3.86 × 10^−8^ ***	0.8561268	0.15259	0.70123	0.0094466
CV 20 ^7^	0.950267	0.34416	0.0560621	121.050994	7.15 × 10^−9^ ***	0.8832551	0.050145	0.82565	0.0031243

^1^ IL-6, Interleukin-6; ^2^ CX3CL1, Fractalkine; ^3^ BDNF, Brain-Derived Neurotrophic Factor; ^4^ PRO 3, Proliferation 3%; ^5^ PRO 20, Proliferation 20%; ^6^ CV 3, Cell Viability 3%; ^7^ CV 20, Cell Viability 20%. *** *p* = 0.001; * *p* = 0.05.

## Data Availability

Data are contained within the article or Appendix A. The original contributions presented in this study are included in the article/Appendix A. Further inquiries can be directed to the corresponding author(s).

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
