# Peer review of "Predicting the Beneficial Effects of Cognitive Stimulation and Transcranial Direct Current Stimulation in Amnestic Mild Cognitive Impairment with Clinical, Inflammation, and Human Microglia Exposed to Serum as Potential Markers: A Double-Blind Placebo-Controlled Randomized Clinical Trial"

_ijms, 2025, doi:10.3390/ijms26041754_

Round 1

Reviewer 1 Report

Comments and Suggestions for Authors

The paper titled Predicting beneficial effects of cognitive stimulation and transcranial direct current stimulation in amnestic mild cognitive impairment with clinical, inflammation, and human microglia

exposed to serum as potential markers: A Double-Blind Placebo-Controlled Randomized Clinical Trial covers an important topic of finding the possible treatment in order to stop progressing to dementia after MCI.

Point 1. The introduction is too long, and because of that, it is difficult to follow. Please summarize the paragraphs in order to gain clarity.

Point 2. The sample size is small. Please explain how the sample size was determined and its efficiency for statistical analysis.

Point 3. Please split the table 2, as it is too large.

Point 4. The resolution of Figure 2 is poor, particularly in Part B. Please correct that.

Point 5. According to the literature, is there any evidence regarding the effect of induced polarised current on the speed of progression to dementia after MCI? 

Reviewer 2 Report

Comments and Suggestions for Authors
  1. The statement "no effective treatment exists" in the abstract is rather vague. Please provide more specific limitations of previous studies to support this claim.
  2. In the introduction, you mention that two systematic reviews and meta-analyses did not distinguish between tDCS alone and tDCS combined with other interventions, making it difficult to determine the specific effects of tDCS. However, your study design does not address this limitation by examining tDCS as a standalone intervention. Please clarify the rationale for your chosen study design given this critique of previous research.
  3. Could you provide justification for choosing 15 sessions as the intervention duration? Is this based on previous literature or other considerations?
  4. While you state that participant selection was based on Petersen's diagnostic criteria, please provide more detailed inclusion and exclusion criteria specific to your study. This will help readers understand exactly how participants were selected.
  5. What was the rationale for selecting the dorsolateral prefrontal cortex (DLPFC) as the stimulation target for tDCS in this study? Please explain the reasoning behind this choice.

Reviewer 3 Report

Comments and Suggestions for Authors

This manuscript investigates the combined effects of cognitive stimulation (CS) and transcranial direct current stimulation (tDCS) on patients with amnestic mild cognitive impairment (aMCI). While the study presents intriguing preliminary findings, several aspects require further clarification and exploration. The observed changes in serum IL-6 and BDNF levels are intriguing. I suggest a discussion of the potential biological mechanisms underlying these changes and their relationship to cognitive function improvements would significantly enhance the manuscript.  Also, a more detailed exploration of the complex interplay between inflammation (as indicated by IL-6 levels) and cognitive function in aMCI would provide valuable insights into the pathophysiology of the disease. Overall, this study presents a promising initial exploration of the combined effects of CS and tDCS in aMCI.

Reviewer 4 Report

Comments and Suggestions for Authors To whom it may concern, Please add these comments to my review: 1. Please state clearly in the introduction the main question/questions of your research and in the conclusion section if the results have confirmed your initial hypotheses. 2. Please clearly state in the introduction why your research is original (and different compared to other similar studies and what brings new to the literature/medical practice. 3. The abbreviation list should be in alphabetical order. 4. A lot of references are outdated. For the articles published before 2015, please find similar newer ones. 5. Please improve the conclusion section. I suggest you use a two-paragraph structure - in the first paragraph, you summarize the results of your research, and in the second paragraph, you should offer 2-3 future research directions/gaps of knowledge that need further study. 

Round 2

Reviewer 2 Report

Comments and Suggestions for Authors

Although this revision does not address all the concerns I raised in the previous review, it does seem to have improved. 

Reviewer 3 Report

Comments and Suggestions for Authors

Dear Authors,

Thank you for submitting your revised manuscript. I have carefully reviewed the changes you have made, and I am pleased to see that you have adequately addressed the suggestions raised during the initial review process.

The manuscript is now well-structured, the arguments are clearly presented, and the supporting evidence is compelling. The revisions have significantly strengthened the paper and improved its overall quality.

Therefore, it is my opinion that the manuscript is now suitable for acceptance in its current form. 

Reviewer 4 Report

Comments and Suggestions for Authors

Dear authors,

You have addressed most of my requests. Still, the abstract is unstructured (make it structured). Please revise figure 2 (page 9).